# Detecting transitions and quantifying differences in two SST datasets using spatial permutation entropy

Juan Gancio<sup>1</sup>, Giulio Tirabassi<sup>2</sup>, Cristina Masoller<sup>1</sup>, and Marcelo Barreiro<sup>3</sup>

Correspondence: Juan Gancio (juan.gancio@upc.edu)

Abstract. Weather prediction systems rely on the vast amounts of data continuously generated by Earth modeling and monitoring systems, and efficient data analysis techniques are needed to track changes and compare datasets. Here we show that a nonlinear quantifier, the spatial permutation entropy (SPE), is useful to characterize spatio-temporal complex data, allowing detailed analysis at different scales. Specifically, we use SPE to analyze ERA5 and NOAA OI v2 sea surface temperature (SST) anomalies in two key regions, Nino3.4 and Gulf Stream. We perform a quantitative comparison of these two SST products and find that SPE detects differences at short spatial scales (<1 degree). We also identify several transitions, including a transition that occurs in 2007 when ERA5 changed its sea–surface boundary condition to OSTIA, in 2013 when OSTIA updated the background error covariances, and in 2021 when NOAA SST changed satellite, from MeteOp–A to MeteOp–C. We also show that these transitions are not detected by standard distance and cross-correlation methods.

## 10 1 Introduction

Due to the large amount of data generated by Earth modeling and monitoring systems, much effort is currently being devoted to developing new, efficient climate data analysis techniques (Dijkstra et al., 2019; Messori et al., 2017; Boers et al., 2019; Gupta et al., 2021; Díaz et al., 2023; Krouma et al., 2024; Allen et al., 2025). Ordinal analysis (Bandt and Pompe, 2002) is a popular symbolic method of time-series analysis that has been applied to geophysical data. For example, ordinal analysis was used to study time series of surface air temperature anomalies in a regular grid over the earth's surface (reanalysis data from the National Center for Environmental Prediction/National Center for Atmospheric Research NCEP/NCAR) and uncovered long-range tele-connections across multiple time scales (Barreiro et al., 2011; Deza et al., 2013). The ordinal method is based on estimating the probabilities of symbols, known as ordinal patterns (OPs), defined in terms of the temporal order of the relative values of L data points. As an example, for L=3, triplets of consecutive data values such that  $x_t < x_{t+1} < x_{t+2}$  are encoded in the symbol "012" where the digits represent the rank of the corresponding value within the triplet. The symbols' probabilities are estimated from their frequencies of occurrence within the time series and their Shannon entropy, known as permutation entropy (PE), is a quantifier of nonlinear temporal correlations. PE is low when some OPs are much more probable than others.

<sup>&</sup>lt;sup>1</sup>Universitat Politècnica de Catalunya, Departament de Física, Rambla Sant Nebridi 22, Terrassa 08222, Barcelona, Spain.

<sup>&</sup>lt;sup>2</sup>Universitat de Girona, Departament de Informàtica, Matemàtica Aplicada i Estadística, Universitat de Girona, Carrer de la Universitat de Girona 6, Girona 17003, Spain.

<sup>&</sup>lt;sup>3</sup>Departamento de Ciencias de la atmósfera y Física de los Océanos, Facultad de Ciencias, Universidad de la República, Montevideo, 11400, Uruguay.

and maximum when all possible OPs are equally probable (Bandt and Pompe, 2002). Ordinal analysis is computationally very efficient and robust to the presence of artifacts and noise. The use of time-lagged (non-consecutive) data points adds versatility to the method, since it allows to select different temporal scales for the analysis. For example, for analyzing a climatic time series with monthly resolution, the L=3 OPs can be defined by considering data values in three consecutive months (e.g., January, February, March; February, March, April; etc), in three consecutive years, or equally spaced over a period of time (for example, a year) (Deza et al., 2013). An important limitation of the ordinal methodology is that the symbolic coding rule does not take into account the actual values of the data points, but their relative values, and therefore, ordinal analysis gives partial information, complementary to that obtained by using standard time series analysis techniques.

Ordinal analysis was originally proposed for time series analysis and adapted for the analysis of gridded two-dimensional spatial data (Ribeiro et al., 2012), by defining the OPs in terms of the relative values of L grid points. Spatial ordinal analysis is a versatile tool because one can choose different "shapes" and/or different spatial orientations for the symbols. For example, for symbols defined in terms of the data values of L=4 grid points, one can consider squares of  $2\times 2$  grid points, a line (horizontal or vertical) of 4 grid points, an "L" composed by 3+1 grid points, etc. Furthermore, the use of spatially lagged grid points allows tuning the spatial scale of the analysis.

The spatial permutation entropy (SPE), which is Shannon's entropy estimated from the probabilities of spatial ordinal patterns, has been used to analyze images, art works and textures (Sigaki et al., 2018, 2019; Tirabassi and Masoller, 2023; Tirabassi et al., 2023; Muñoz-Guillermo, 2023; Tarozo et al., 2025). It has also been used to analyze complex spatio-temporal data such as EEG recordings (Boaretto et al., 2023; Gancio et al., 2024) and cardiac synthetic data (Schlemmer et al., 2015, 2018). However, to our knowledge, SPE has not yet been tested on climate data.

Since SPE can be calculated from the relative values of a climate variable at a given time in a particular geographic region, it yields information about nonlinear spatial correlations of that climate variable, in that region, at that time. In contrast, the "temporal" PE of the variable at a particular grid point is calculated from the analysis the variable's time series at that grid point, and therefore, it yields information about nonlinear temporal correlations of that variable, at that grid point.

Our goal is to demonstrate that SPE is a reliable and versatile tool, and specifically, is able to capture subtle differences between datasets and also, changes within the same dataset. We focus on a key variable, sea surface temperature (SST) anomalies, and compare two SST products, ERA5 and NOAA Optimal Interpolation version 2 (NOAA OI v2), in two key regions, the equatorial Pacific and the the Gulf Stream. We show that SPE identifies differences in the datasets in short spatial scales, which can be more or less pronounced over different periods of time. We interpret our findings in terms of changes in the methodologies and data used to construct the SST products.

#### 2 Data

We consider monthly SST anomalies in El Niño3.4 region (170W–120W, 5N–5S), and in the western north Atlantic (32.5N–42.5N, 67.5W–45W), a box centered on the Gulf Stream (see Fig. 1a). We analyze NOAA Optimal Interpolation version 2 (NOAA OI v2) (Reynolds et al., 2007; Huang et al., 2021), and ERA5 global reanalysis (Hersbach et al., 2020). Both datasets

**Figure 1.** Panel (a) highlights the regions of interest: Niño3.4 (in green), and the Gulf Stream (in orange). Panels (b) and (c) show the SST anomaly in the Niño3.4 region, and panels (d) and (e), in the Gulf Stream region, calculated from ERA5 (b, d) and NOAA OI v2 (c, e) datasets. In panels (b)–(e), the thick lines represent the spatial mean of the anomalies, while the shading indicates the spatial standard deviation.

have spatial resolution of  $0.25^{\circ} \times 0.25^{\circ}$ . ERA5 starts in January 1940, while NOAA OI v2 starts in September 1981; both extend to June 2025 (therefore, the NOAA time series have 526 datapoints each, while the ERA5 time series have 1026 datapoints each).

NOAA SST includes observations from ships, drifting and moored buoys, and the Advanced Very High Resolution Radiometer (AVHRR) (Huang et al., 2021) retrieved from NOAA series and MetOp-A/-B satellites by U.S Navy before November 2021. After this date, NOAA SST switched to the Advanced Clear Sky Processor for Ocean (ACSPO) (Huang et al., 2023; Jonasson et al., 2020) satellite SSTs retrieved from AVHRR and the Visible Infrared Imager Radiometer Suite (VIIRS) (Huang et al., 2023).

ERA5 SST is the combination of HadISST2 (Titchner and Rayner, 2014) up to August 2007 and OSTIA (Donlon et al., 2012) from September 2007 onwards (Hirahara et al., 2016). HadISST2 assimilates in-situ observations as well as two radiometers: AVHRR and the Along Track Scanning Radiometer (ATSR).

OSTIA was originally constructed at a resolution of  $0.05^{\circ}$  and includes in situ data from various sources, as well as derived from several satellite products including AVHRR and VIIRS. It is worth noting that the higher resolution of OSTIA allows it to better resolve the tropical instability waves and sub-mesoscale eddies in the midlatitudes (Hirahara et al., 2016).

Within the regions of interest (see Fig. 1a), both datasets employ a similar grid (with  $40 \times 200$  grid points for the Niño3.4, and  $40 \times 90$  for the Gulf Stream region), the only difference being a small offset of  $0.005^o$  both in latitude and longitude.

## 3 Analysis tools

#### 3.1 Ordinal patterns and spatial permutation entropy

Ordinal analysis is a symbolic data analysis technique proposed by Bandt and Pompe (2002) that has been extensively applied in a wide variety of different scientific fields (Leyva et al., 2022). Ordinal analysis takes an ordered series of values that represents the evolution of a certain variable (usually known as time series),  $x_t$ , where  $x_t \in \mathbb{R}$ , and  $t \in [1, ..., N]$ , and translates it into a sequence of symbols:  $s_t$ . This operation requires only two parameters: the symbol length, L, and a lag,  $\delta$ . These parameters are used to select a sequence of L data points  $[x_t, x_{t+\delta}, x_{t+2\delta}, ..., x_{t+(L-1)\delta}]$ , which is assigned the symbol  $s_t = [\pi(t), \pi(t+\delta), ..., \pi(t+(L-1)\delta)]$  (known as ordinal pattern) where  $\pi(\cdot)$  is the permutation index that sorts the selection in ascending order:  $x_{\pi(t)} \le x_{\pi(t+\delta)} \le ... \le x_{\pi(t+(L-1)\delta)}$ . As an example, if the symbols are defined in terms of three consecutive data points  $(L=3 \text{ and } \delta=1)$ , the data points (3.2, 4.4, 1.3) are represented by the ordinal pattern "201", while (1.3, 3.2, 4.4) are represented by the pattern "012" and (4.4, 3.2, 1.3) by the pattern "210", etc. This operation is repeated for every  $t=1, ..., n=N-(L-1)\delta$ , producing a sequence of n ordinal patterns. Note that the number of possible patterns is L!, which imposes a limit to the maximum length of the patterns because the length of the sequence of symbolics, n, needs to be much larger than the number of possible patterns, L!, in order to obtain a reliable estimation of symbols' probabilities. In practical terms, typical values of L range from 3 to 6.

If  $n_i$  is the number of times that the symbol i appears in the symbolic sequence, then the probability of the symbol is estimated as  $p_i = n_i/n$ , with  $i \in [1, ..., L!]$ . The set of probabilities are then used to calculate the Shannon's entropy,

$$H = -\frac{1}{\log(L!)} \sum_{j=1}^{L!} p_j \log(p_j). \tag{1}$$

known as permutation entropy (PE). The coefficient  $1/\log(L!)$  normalizes H between 0 and 1, enabling the comparison between values extracted from ordinal pattens of different lengths. Values of H close to 0 represent time series where a single symbol predominates (such as periodic or trending series), whereas high values of H occur when all symbols are equally probable, typically indicating fully stochastic time series.

# 3.2 Symbol orientations in 2D spatio-temporal gridded data

The SST anomalies in the two regions of interest are represented as the time evolution of 2-dimensional  $N \times M$  gridded datasets,  $X_{i,j}(t)$  with  $i \in \{1,\ldots,N\}$ ,  $j \in \{1,\ldots,M\}$ ,  $t \in \{1,\ldots,T\}$ , where the index i corresponds to different latitudes, the index j to different longitudes, and T is the number of time steps in the series. Given the parameters L (symbol length) and  $\delta$  (spatial lag), at each time step t we encode the data into spatial OPs using two possible symbols: those in the North-South (NS) direction, defined from the values, at time t, along the columns of the grid,  $\{X_{i,j}(t), X_{i+\delta,j}(t), \ldots, X_{i+\delta(L-1),j}(t)\}$ , and those in the East-West (EW) direction, defined from the values, at time t, along the rows,  $\{X_{i,j}(t), X_{i,j+\delta}(t), \ldots, X_{i,j+\delta(L-1)}(t)\}$ , see Fig. 2. In both cases there are L! possible patterns and from their probabilities at time t,  $p_j(t)$ , we compute the spatial permutation entropies at time t,  $H_{NS}(t)$  and  $H_{WE}(t)$ , using Eq. (1).  $H_{NS}(t)$  and  $H_{WE}(t)$  will be close to 1 when there is no

| 1.1  | 3.2  | 0.7  | 4.4  | -3.8 | 1.3 |
|------|------|------|------|------|-----|
| -1.4 | 0.1  | -1.9 | 2.7  | 3.1  | 1.4 |
| 4.1  | -3.2 | 1.0  | -2.4 | -0.3 | 0.6 |

Figure 2. Illustration of the procedure used to define symbols, known as ordinal patterns. The  $6 \times 3$  grid represents the values of SST anomalies (ERA5 or NOAA) at time t, in a geographical region. Two examples of ordinal patterns (OPs) of length L=3 are presented: The values marked in green in the NS direction (1.1, -1.4, 4.1) have a spatial lag  $\delta=1$  and give pattern "102", while the values marked in orange in the WE direction (3.2, 4.4, 1.3) have a spatial lag  $\delta=2$  and give the pattern "120". The spatial permutation entropies at time t,  $H_{NS}(t)$  and  $H_{WE}(t)$ , are calculated from the probabilities of OPs that are defined by SST anomalies at grid points along the NS and WE directions respectively. By considering different  $\delta$  values we are able to tune the spatial scale of the analysis.

**Table 1.** Number of symbols (ordinal patterns) defined in the two regions analyzed, for each orientation, and for each spatial lag ( $\delta$ ) considered.

|             | El Niño region (170W–120W, 5N–5S) covers $40 \times 200$ grid points |              |              |              | Gulf Stream region (32.5N–42.5N, 67.5W–45W) covers $40 \times 90$ grid points |              |              |              |
|-------------|----------------------------------------------------------------------|--------------|--------------|--------------|-------------------------------------------------------------------------------|--------------|--------------|--------------|
|             | $\delta = 1$                                                         | $\delta = 2$ | $\delta = 4$ | $\delta = 8$ | $\delta = 1$                                                                  | $\delta = 2$ | $\delta = 4$ | $\delta = 8$ |
| West-East   | 7880                                                                 | 7760         | 7520         | 7040         | 3480                                                                          | 3360         | 3120         | 2640         |
| North-South | 7400                                                                 | 6800         | 5600         | 3200         | 3330                                                                          | 3060         | 2520         | 1440         |

spatial order in the data (all OPs are equally probable), and will be < 1 if the data presents gradients (in the NS or in the WE direction) that make some spatial OPs more or less probable. Since SST values vary over time,  $H_{NS}$  and  $H_{WE}$  will also vary over time.

In this work, we consider symbols defined by L=4 grid points and a spatial lag up to  $\delta=8$ , as these are the largest values that allow us, given the size of the two regions studied, to estimate with good statistics the probabilities of the L!=24 possible patterns. Table 1 shows the number of symbols (defined by L=4 grid points in the geographical region analyzed) when a spatial lag of  $\delta=1$ , 2, 4 and 8 is used. We see that the lowest number (in the Gulf stream with  $\delta=8$ ) is >>24.

# 110 3.3 Distance and cross-correlations measures used to compare SST-ERA5 and SST-NOAA

In this work we demonstrate that spatial permutation entropy can detect differences in ERA5 and NOAA SST products, and an important question is whether such differences can also be detected using standard correlation or distance measures. Therefore, we also compare SST anomaly values using the Average Absolute Difference (AAD), the Pearson's spatial cross-correlation coefficient (r), and the Spatial Mutual Information (SMI), which is a non-linear cross-correlation measure (Celik, 2016; Kumar and Bhandari, 2022).

The Average Absolute Difference is defined as:

115

$$AAD(t) = \langle |X_{i,j}(t) - Y_{i,j}(t)| \rangle_{i,j}, \tag{2}$$

where  $X_{i,j}(t)$  and  $Y_{i,j}(t)$  represent the two gridded datasets and  $\langle \cdot \rangle_{i,j}$  represents the spatial average in the analyzed region, at time t.

120 Pearson's spatial cross-correlation coefficient is defined as:

$$r(t) = \frac{\sigma_{X,Y}(t)}{\sigma_X(t)\sigma_Y(t)},\tag{3}$$

where  $\sigma_X(t)$ ,  $\sigma_Y(t)$  and  $\sigma_{X,Y}(t)$  are the spatial standard deviations and the spatial covariance of  $X_{i,j}(t)$  and  $Y_{i,j}(t)$  at time t. The spatial mutual information (SMI) of  $X_{i,j}(t)$  and  $Y_{i,j}(t)$  at time t is defined as:

$$SMI(t) = H_X(t) + H_Y(t) - H_{X,Y}(t).$$
 (4)

Here  $H_X$  and  $H_Y$  are the entropies of  $X_{i,j}(t)$  and  $Y_{i,j}(t)$  at time t, and  $H_{X,Y}$  is their joint entropy at time t. Depending on which probability distributions are used to calculate the entropies, different values of SMI are obtained. In this work we have considered three different approaches to estimate the probabilities: the probabilities of spatial OPs at each time, that is, OPs defined over gridded points with NS or WE orientation ( $SMI_{NS}$  and  $SMI_{WE}$ ), as well as the probabilities estimated from the histograms of SST anomaly values at each time ( $SMI_{hist}$ ). Since for L=4 OPs there are L!=24 possible patterns, to compute  $SMI_{hist}$  we employed histograms of 24 bins, so that the three SMI values are calculated from probabilities defined over the same number of bins.

#### 4 Results

## 4.1 Detecting transitions with spatial permutation entropy analysis

Figure 3 displays the temporal evolution of the spatial permutation entropy, calculated from the probabilities of L=4 OPs defined from the values of SST anomalies in neighboring grid point (i.e., the spatial lag is  $\delta=1$ ). Panels (a) and (c) display  $H_{NS}$  for the two datasets analyzed, ERA5 in blue and NOAA OI v2 in red, in the two regions analyzed: panel (a) corresponds to the Niño3.4 region, and panel (c), to the Gulf Stream region. Results are presented from the beginning of the datasets (1940 for ERA5 and 1981 for NOAA OI v2). Panels (b) and (d), instead, display  $H_{WE}$ , also in the two regions and for the two datasets under analysis.

In the Niño3.4 region, panel (a) shows that the evolution of  $H_{NS}$  for ERA5 and NOAA OI v2 is quite similar. However, panel (b) shows differences in the evolution of  $H_{WE}$  which persist until 2007. After 2007 the differences reduce until 2022, when the two values of  $H_{WE}$  diverge again. The years when these transitions occur coincide with the switch of the sea-surface boundary condition of ERA5, from HadISST2 to OSTIA, in 2007 (Hersbach et al., 2020), and with the inclusion of MeteOp-C satellite data in NOAA's dataset in November 2021 (Jonasson et al., 2020).

To objectively identify change points in the temporal evolution of the entropies (and in all the quantifiers used), we employed a well-known unsupervised algorithm, the Pruned Exact Linear Time (PELT) (Killick et al., 2012) (see Appendix A for details). PELT has been used to analyze geophysical time series such as temperature (Khapalova et al., 2018), vegetation (Wang and Fan, 2021), and stream flow (Rocha and de Souza Filho, 2020). The arrows in Fig. 3 indicate the change points detected by the

155

Figure 3. Entropies calculated with spatial lag  $\delta = 1$  in (a),(b) Niño3.4 region; (c),(d) Gulf Stream region. Blue lines correspond to the entropy from the ERA5 dataset, red lines to the entropy from the NOAA OI v2 dataset, and the black arrows indicate the change points detected by the PELT algorithm in the ERA5 dataset.

PELT algorithm. We note that no change point is detected in panel (a), but a change point is detected in panel (b), in 2007. It is important to note that the PELT algorithm has a penalty parameter, P, which must be carefully selected to detect genuine changes. When setting P too low, PELT returns too many change points, while when setting P too high, PELT may not detect some change points. The procedure followed to select P is described in detail in Appendix A and is based on two steps: the first consists of the analysis of surrogate time series where the null hypothesis (NH) is that there are no change points, and the second step is the analysis of the robustness of the change points passing the surrogate test, with respect to variations in the penalty parameter. However, we note that NH can sometimes fail (because surrogate signals may have change points) and therefore the selected significance threshold may be too high, which can result in genuine change points not being detected, even if they are visually evident.

In the Gulf Stream region, panel (c) shows differences between the values of  $H_{NS}$  of ERA5 and NOAA OI v2, which become relatively small after 2013, a fact that could be due to the update of the background error covariances in OSTIA in January 2013 (Good et al., 2020; Roberts-Jones et al., 2016). Panel (d) also shows considerable differences between the values of  $H_{WE}$ , from 2021 onward, and which could be due to the inclusion of MetOp-C AVHRR data in NOAA OI v2 (Huang et al.,

Figure 4. Entropies calculated with  $\delta=8$  in (a), (b) Niño3.4 region; (c), (d) Gulf Stream region. Dashed blue lines indicate linear fits of the entropy signals for the ERA5 dataset. A better agreement between NOAA OI v2 and ERA5 signals is observed with  $\delta=8$  than with  $\delta=1$  (Fig. 3). The negative trend in (b) can be due to the strengthening westward of large-scale gradient in the Niño3.4 region, while the positive trends in (c) and (d) reveal the decrease of spatial structures.

2023). While change points in 2007 and 2013 are returned by the PELT algorithm, the one in 2021, although significant in the NOAA signal, does not pass the robustness test (See Table. 1 in Appendix A).

Summarizing, most changes that can be identified by visual inspection are consistent with changes returned by the PELT algorithm when used to analyze  $H_{NS}(ERA5)$ ,  $H_{NS}(NOAA OI v2)$ ,  $H_{WE}(ERA5)$  and  $H_{WE}(NOAA OI v2)$  in Niño3.4 and Gulf Stream regions. The PELT change points are:

- 1. A first transition occurs in ERA5 in 2007, which we interpreted as due to the change of the sea-surface boundary conditions: the inclusion of OSTIA in 2007. This transition is associated to change points that are identified in  $H_{NS}$  and  $H_{WE}$  in the Gulf Stream region, and in  $H_{WE}$  in the Niño3.4 region, but not in  $H_{NS}$  in the Niño3.4 region.
- 2. A second transition occurs in ERA5 in 2013, which we interpreted as due to the update of OSTIA. This transition is associated to change points that are identified in the Gulf stream region, both in  $H_{NS}$  and  $H_{WE}$ , but not in the Niño3.4 region. Figure 4 displays the entropies as in Fig. 3, but now the ordinal patterns are defined with non-neighboring grid points. Specifically, the grid points are spaced by a lag  $\delta = 8$  that corresponds to  $2^{\circ}$ . Now we see that the temporal evolution of the

Figure 5. Analysis of El Niño region, in the time interval in which ERA5 and NOAA OI v2 datasets are available (1981-2025). Panels (a)-(d) display the temporal evolution of the SST anomaly (black line, right vertical axis) and the temporal evolution of the spatial permutation entropy (red and blue, left vertical axis) calculated with ordinal patterns with NS (a, c) and WE (b, d) orientation. In (a), (b) the spatial lag used to define the ordinal patterns is  $\delta=1$ , in (c), (d)  $\delta=8$ . In panel (a), where the OPs are aligned along the NS direction, the entropy signals are negatively correlated with the SST anomaly (the Pearson cross-correlation coefficient is r=-0.23 for NOAA, and r=-0.20 for ERA5), while in panel (b), where the OPs aligned along the WE direction, the entropy signals and the SST anomaly are positively correlated (r=0.42 for NOAA, and r=0.37 for ERA5). In panels (c) and (d) the correlations are, in the NS direction: r=-0.32 for NOAA and r=-0.16 for ERA5; in the WE direction: r=-0.04 for NOAA and r=-0.11 for ERA5.

entropies of ERA5 and NOAA OI v2 is consistent, in the two regions and for the two orientations. Discrepancies are small: not only the fluctuations are highly correlated, but also, the trends are similar, but no change points are detected at this scale. For instance, in panels (a) and (b) we see that in the Nino3.4 region, the entropies from the two datasets display very similar trends. The trends are negative and more pronounced in  $H_{WE}$ . These trends are interpreted as due to the long-term SST variations over the equatorial Pacific, since over the years, SST has warmed in the west and cooled in the east (Wills et al., 2022). This represents a westward large-scale gradient over the Niño3.4 region that can make the patterns that encode monotonic increasing or decreasing relative ordering of data values, such as "0123" and "3210", more prevalent, thus decreasing the entropy.

In the Gulf Stream region, which is also heating due to global warming (Seidov et al., 2017; Todd and Ren, 2023),  $H_{WE}$  and  $H_{NS}$  in ERA5 and in NOAA present a positive trend, which reveals a decrease of spatial structure, as it means that the different

symbols become equally probable. In this case, as the northwest part is warming faster than the rest of this area (Bulgin et al., 2020), it decreases the climatological SST gradient across the Gulf stream, thus homogenizing the entire region.

Figure 5 shows the same signals as Fig. 3 and Fig. 4 for the Niño3.4 region, but in the period 1981–2025, and the mean SST anomaly of this region. In panel (b), we can see that  $H_{WE}$  with  $\delta=1$  from the NOAA dataset is positively correlated with the SST anomaly (the Pearson cross-correlation coefficient is r=0.42), most clearly between 2007 and 2022 (r=0.70). The  $H_{WE}$  values are the largest during El Niño years, capturing the decrease in SST gradients along the longitudinal direction. For ERA5, this correlation is strongest from 2007 onward (r=0.57). In contrast,  $H_{NS}$  with  $\delta=1$  (panel (a)) is negatively correlated with the SST anomaly (r=-0.23 for NOAA, and r=-0.20 for ERA5), thus during some El Niño events (such as 1997-1998 and 2015-2016 years) the entropy decreases, reflecting the north-south gradients that occur as the equatorial zone is warmer than the north-south edges of the region. This is also observed for some La Niña events, such as the one in 1988-1989 or in 1998-1999.

For the same analysis but with  $\delta=8$ , sudden drops of  $H_{WE}$  (Fig 5d) can be observed, and some correlate with El Niño events, as the ones in 1982–1983, or 1997–1998. This represents the opposite behavior as reported for the  $\delta=1$  scale, which is expected as the spatial structures that are formed by the uneven warming/cooling of the region during ENSO events, which are accentuated due to global warming (Cai et al., 2014; Xie et al., 2010), appear at these larger scales, while at the smaller scale the variations of SST are more uniform. Some drops are also seen in the temporal evolution of  $H_{NS}$  (Fig 5c), although at this scale ( $\delta=8$  is equivalent to each "word" spanning  $6^o$ )  $H_{NS}$  cannot capture the NS gradients appropriately. This problem does not occur in the longitudinal direction, as the gradients in this direction have a larger scale.

# 4.2 Comparison between ERA5 and NOAA datasets

To analyze similarities and differences between ERA5 and NOAA, in this section we only consider the period when both datasets are available (1981–2024). To visualize the effect of the spatial lag between the grid points, Fig. 6 displays the entropies (as done in Figs. 3 and 4), for grid points that are consecutive (i.e., they are separated by  $0.25^o$ , that is,  $\delta = 1$ ), that are separated by  $0.5^o$  ( $\delta = 2$ ), by  $1^o$  ( $\delta = 4$ ), and by  $2^o$  ( $\delta = 8$ ). Therefore, in this figure, the left and write columns display the same entropies as in Figs. 3 and 4.

We observe that as  $\delta$  increases the behavior of  $H_{NS}$  and  $H_{WE}$  for the two datasets converge, which indicates that the differences found between ERA5 and NOAA occur mainly at short spatial scales. We also notice that, as  $\delta$  increases, so do the entropies, which suggests that the gradients become less pronounced as the spatial scale increases.

To perform a quantitative comparison between the datasets, we begin by analyzing their SMI, reported in Fig.7. In panels (a–h) we report SMI estimated from ordinal probabilities, in particular  $\delta = 1$  for panels (a–d) and  $\delta = 8$  for panels (e–h), while panels (i) and (j) display SMI estimated from the probabilities of values of SST anomalies.

When SMI is calculated from the ordinal probabilities, in the two regions and for the two orientations, we can see that it increases with time, which reveals that the amount of information shared by the datasets increases with time. PELT detects a transition in 2007 in all the panels.

Figure 6. Effect of the lag between grid points. Panels (a)–(h) correspond to the Niño3.4 region, and (i)–(p) to the Gulf Stream region. We can see that as  $\delta$  increases, the entropy differences gradually decrease, revealing that the SPE analysis captures the differences between ERA5 and NOAA datasets at small spatial scales.

For el Niño 3.4 region, we observe that the PELT analysis now detects a transition at the small scale ( $\delta=1$ ) in the NS orientation (Fig. 7c), which was not detected in the analysis of the  $H_{NS}$  signal at this scale (Fig.3). We note that El Niño events have an impact on  $SMI_{WE}$ , as we observe that the coherence between the datasets decreases during such events, as we can observe, for example, during the ElNiño events of 1997, 2010, and 2015. We highlight that this effect is observed at all scales, and in the current version of the products, since it is observed during the 2023 warm ENSO event. Regarding  $SMI_{hist}$ . (Fig. 7e), although we observe some increase in the average value in 2007 and 2015, no robust and significant change points are detected in this signal.

For the Gulf Stream region (Fig. 7b, d, f, h, and j), all the SMI values reported present oscillations around a relative constant value before 2007, which appears to increase consistently from this year on. Additionally, on the large scale, we detected two other transitions: one at end of 2015/beginning 2016 (the same observed in  $SMI_{NS}$  for  $\delta=1$  in ElNiño region), and another one in 2021. Both transitions correlate with changes in the satellites experienced by NOAA SST those years (Huang et al., 2023). We also note that 2016 corresponds to the start date of version 2.1 of the NOAA OI product, which also includes the

Figure 7. Spatial mutual information, Eq. (4), between SST anomalies of ERA5 and NOAA datasets. The left column corresponds to the Niño3.4 region, while the right column, to the Gulf Stream region. In the first three rows, panels (a–h), SMI is calculated from the probabilities of ordinal patterns with  $\delta=1$  (a–d) and  $\delta=8$  (e–h); the OPs are constructed with WE orientation in (a), (b), (e), and (f), and with NS orientation in (c), (d), (g) and (h). In the bottom row, panels (i) and (j), SMI is calculated from the probabilities of the data values, using the same number of bins, 24, as the number of possible OPs. In all cases SMI increases with time, revealing, as expected, that the discrepancies between ERA5 and NOAA datasets diminish; however, note the difference in the vertical scales in (a–d) and (e–h): The higher SMI values when ordinal patterns are defined with a spatial lag  $\delta=8$  relative to  $\delta=1$  reveal that the agreement between the two data sets is better at long spatial scales than at short ones. The arrows indicate change points detected by the PELT algorithm.

**Figure 8.** Average absolute difference, AAD (panels a and b), and spatial Pearson's correlation coefficient, r (panels c and d),, between ERA5 and NOAA datasets in (a), (c) Niño3.4 region, and (b), (d) Gulf Stream region.

Argo observational data (Huang et al., 2021), and several changes in the conventional and radiance observations assimilated by ERA5 also occurred that year (Hersbach et al., 2020).

Regarding  $SMI_{hist.}$ , we observe that the annual cycle has some impact on this signal, but not on  $SMI_{NS}$ , nor  $SMI_{WE}$ .

Finally, to demonstrate the added value of using symbolic ordinal analysis, in Fig. 8 we present the results obtained with two well-known measures of linear relationship between two datasets: the average absolute difference, AAD, and the spatial Pearson's correlation coefficient, r. They provide complementary information because when AAD and r are both high, the data sets differ in values but their spatial distributions are consistent, whereas when both are low, there is agreement between the values, but not in the spatial distributions. In Fig. 8 we see that both measures show continuous improvement of the agreement between the two datasets in the two regions; however, there are oscillations and no clear transitions are observed. In the Gulf Stream region AAD and r show a strong coupling with the annual cycle, with disagreement (high AAD and low r) peaking during northern winters. Increased cloud coverage in this region during winters could difficult infrared measurements, which may lead to larger differences between ERA5 and NOAA datasets. On the other hand, in Niño3.4 region ENSO events affect the agreement between the datasets, but their effect is captured differently by the two measures. For example, AAD peaks during El Niño in 1988, 1997 and during La Niña of 2011, but these events do not affect r, and vice-versa, La Niña in 1996 and 1999, and El Niño in 2004, affect r but not AAD.

Regarding the CPD analysis, no robust and significant change points are detected in these signals.

#### 4.3 Summary and robustness of detected points

Because PELT assumes that the data is piecewise-stationary, and that within each segment the distribution follows an independent and identically distributed sequence of random variables (Garreau and Arlot, 2018), signals that show a linear trend

like the ones in Figs. 3 and 4, must be detrended to avoid the detection of spurious change points. We ensured the robustness of our change point detection by detrending and analyzing separately segments of the time series displaying different trends. Therefore, the change points detected by our analysis corresponds generally to abrupt variations in the time series statistics. "Change" in other senses could also be studied, for example, changes of slope in the linear trends; however, this is beyond the scope of this study, as our goal is to demonstrate the capability and flexibility of the spatial ordinal analysis methodology, and to analyze the most robust changes detected.

The signals of SMI shown in Fig. 7, especially those obtained from ordinal patterns (panels a-h), present sections with and without a linear trend, most notably pre- and post-2007. In these cases, we run the PELT algorithm without any pre-processing, which provides us with an initial segmentation of the signal. Then, we statistically tested for the presence of a linear trend in each of these segments of this initial partition, using a Wald test. If two continuous segments do not present a significant linear trend, or only one of them does, the original change point is considered genuine. Otherwise, i.e., both segments present a significant linear trend, the change point is considered induced by this trend, and both segments have to be detrendend before running the PELT algorithm again on them. All the change points found on the small scale (Fig. 7(a-d)) correspond to a transition between the signal having a significant nonzero linear trend and a nonsignificant linear trend (or vice versa). On the other hand, the 2007 change point at the large scale (Fig. 7(e-h)) always present a linear trend before and after this year, but these change point persist after detrending the signal. Regarding the other change points detected at the large scale in the Gulf Stream region, the one in 2016 in  $SMI_{WE}$  and the one in 2021 in  $SMI_{NS}$  correspond again to transitions trend/no-trend, while the one in 2021 in  $SMI_{WE}$  corresponds to a jump: no significant linear trend before or after.

For  $SMI_{hist.}$  in the Gulf Stream region region (Fig. 7j, the algorithm detects two change points, but one in 2007 and the other one in 2021. The first one corresponding to a change point with linear trend before and after, which survives detrending, and the second one to a trend/no-trend transition.

In addition to the suitability of the chosen surrogates mentioned before (change points still occur in the surrogates, which increases the significance threshold and prevents the detections of true change points), we highlight that the last step in our CPD analysis, the robustness check of the significant change points, allowed us to distinguish between what look like spurious detections and evident ones (such as 2007). But some of the discarded change points (as in 2013 or 2021 in  $SMI_{WE}$  and  $SMI_{NS}$  for  $\delta=1$  in the Gulf region), which are visually evident and significant, do not present robustness levels larger than the defined threshold. In fact, although the relative robustness (R) is computed from each signal, the threshold for significant robustness (R) is computed from the robustness distribution obtained by considering all the robustness values from all the change points detected in all the time series (entropies, mutual information, and linear measures, see Appendix A). Therefore, the robustness values of a single time series affect the robustness threshold by affecting the median of the distribution. For example, a time series with many robust change points (high P) would increase the median of P, making P0 small and the robustness of its own change points less significant; or a time series with a single very robust change points (high P1) increases the overall median of P1 potentially making non-significant robust change points of another time series. However, the change points that pass the double check of significance and significant robustness are simultaneously robust to other change points in the signals and to all other significant change points.

While ordinal analysis allows us to detect several changes that could not be detected by the distance or cross-correlation measures considered, it has the drawback that ordinal analysis typically has computational costs and hyper-parameters (the length of the symbol and the lag) which need to be carefully selected. However, ordinal analysis has the important advantage of offering a large degree of flexibility, by allowing to tune the shape of the pattern, and the spatial scale of the analysis.

We have shown that a nonlinear quantifier, the spatial permutation entropy (SPE), is useful and flexible for analyzing the

#### 5 Conclusions

spatiotemporal dynamics of SST anomalies. SPE is computed from the probabilities of symbols, known as ordinal patterns (OPs), defined by four SST anomaly values in grid points that are geographically oriented north-south (NS) or west-east (WE), and that are either neighboring grid points or separated by a spatial lag,  $\delta$ . We used SPE to analyze ERA5 and NOAA OI v2 SST anomalies in two key regions: Niño3.4 and the Gulf Stream. Temporal variations of SPE allow for detailed model intercomparison: We found remarkable similarities in ERA5 and in NOAA OI v2 when the spatial analysis scale is long enough (when the OPs are defined by SST anomalies in grid points separated by  $2^{\circ}$ ), but we uncovered differences between ERA5 and NOAA OI v2 when the spatial analysis scale is short (when the OPs are defined by SST anomalies in neighboring grid points). In addition, SPE temporal variation allowed us to identify four transitions that occur in 2007 when ERA5 changed its sea-surface boundary condition to OSTIA, in 2013 when OSTIA updated the background error covariances, in 2016 when several changes occurred both in NOAA (as the inclussion of MeteOp-B and Argo data) and ERA5, and in 2021 when NOAA SST changed satellite observations from MeteOp-A/-B to MeteOp-C. While these transitions can be observed by simply inspecting the temporal evolution of SPE, to corroborate these findings we applied an unsupervised change detection algorithm. These differences add on to previously reported discrepancies (Yao et al., 2021; Dai, 2023). We also report different SPE trends on the equatorial Pacific and Gulf Stream regions in the last decades, which are consistent with different responses to greenhouse gas forcing (uneven warming/cooling). Although these long-term changes in trends are known and can be observed with other analysis tools, the localized, well-defined change points found here were not detected by cross-correlation

For future work, it will be interesting to integrate in the definition of the OPs not only the SST spatial variation (as done here) but also the temporal variation. This can be achieved by computing OPs in time, or by building the OPs in a way they extend both in space and time. The introduction of temporal OPs will allow us to integrate information from different time scales, since varying the value of  $\delta$  allows us to study daily, intra-seasonal or interannual variations in SST.

and mutual information analysis, and to the best of our knowledge, they have not been reported previously.

The increase in the spatial resolution of SST products seen in latest years will allow to investigate the role of the mesoscale and sub-mesoscale dynamics on bio-geo-chemical processes such as the importance of eddies and fronts on the distribution of nutrients, phytoplankton growth and carbon uptake. Thus, it is crucial that different SST products are able to characterize the small scale SST variability adequately. Our analysis reveals differences in SST datasets at these scales that could hinder their use. However, it also reveals clear improvements in the similarity of SST datasets in recent years, following the introduction of

340

new satellite observations and advanced data processing methodologies, confirming significant advances in Earth observation systems.

Code and data availability. NOAA OI v2 data was obtained from the KNMI Climate Explorer (https://climexp.knmi.nl/select.cgi?id=someone@somewhere&field=sstoiv2\_monthly\_mean). ERA5 data was obtained from the Copernicus Climate Data Store (https://cds.climate.copernicus.eu/datasets/reanalysis-era5-single-levels-monthly-means?tab=overview). The code used for the analysis and the generation of the figures is available at https://github.com/juangancio/climate-spatial-analysis, and archived at https://doi.org/10.5281/zenodo.17250157.

## 320 Appendix A: Unsupervised change point detection (CPD) algorithm PELT

To formally detect transitions in the quantities introduced in Sec. 3.1, we applied a change point detection (CPD) algorithm known as Pruned Exact Linear Time (PELT) (Killick et al., 2012) implemented in the Python package ruptures (Truong et al., 2020), with a Gaussian kernel cost function. Given a penalty parameter, P, PELT returns a number of change points. The value of P needs to be carefully selected because it determines the number of change points found: if it is too low, too many change points are detected, while if it is too high, no change point is detected.

CPD analysis typically assumes that the signals are piecewise stationary (Truong et al., 2020), therefore linear trends have to be removed before using the PELT algorithm. This was the case for the time series of the spatial permutation entropy (Eq. 1), and of the linear measures (ADD and r), where a simple detrend was sufficient as pre-processing. In contrast, for the time series of the spatial mutual information (Eq. 4), sections with and without a linear trend are observed in the signal; hence we performed the following sequence of steps: 1) The time series were first divided in segments where the linear trend is constant. To do this, we combined the PELT algorithm with a statistical test of the linear trend: we run PELT without detrending the input, and test if there is a significant (p-value 

Figure A1. Robustness of the change points detected in the signal of  $H_{WE}$  from the ERA5 dataset with  $\delta = 1$  in the ElNiño3.4 region. Panel (a) shows the original signal detrended, while panel (b) shows its iAAFT surrogate (Schreiber and Schmitz, 1996). Panels (c) and (d) display the corresponding bar plots showing the persistence of each detected change point as the penalty parameter is increased. Dashed line corresponds to  $\overline{P^*}$  of the signal, while the continuous line marks the 99.5th percentile of  $P^*$  from the surrogates.

After detecting the significant change points, we tested their robustness, relative to other (including not significant) change points picked up by the algorithm. For this, we first calculate the maximum penalty parameter for which each change point is detected,  $P^*$ , and then calculate the relative difference between  $P^*(t)$  of the significant change point that occurs at time t and the median of  $P^*$  for all the change points detected in the signal  $(\overline{P^*})$ :

$$R(t) = \frac{P^*(t) - \overline{P^*}}{\overline{P^*}}.$$
(A1)

We selected the median of the distribution of R obtained from every significant change point (see Table 1),  $\overline{R} = 19$ , as the threshold, such as if  $R \ge 19$  the change point is considered robust.

Now, in Fig. A1c, we show an example of such analysis performed in the signal of  $H_{WE}$  from the ERA5 dataset with  $\delta=1$  in the ElNiño3.4 region (blue signal en Fig. 3a). Here we show the change points detected as a function of the penalty parameter, P, in cyan. In black continuous line, we mark the 99.5th percentile of the distribution of maximum P for which each change point is detected,  $P^*$ , in the surrogated signals. In black dashed line, we display the median of  $P^*$  from the signal:  $\overline{P^*}$ . In red we have highlighted the  $P^*$  of the two significant change points (those which can be detected for P larger than the

Figure A2. Same as Fig. A1, but for the Gulf Stream region.

95th percentile of  $P^*$  from the surrogated signals), but only one (in 2007) is robust enough (the relative distance, R, between  $P^*$  and  $\overline{P^*}$  is larger than 19) to be considered a true change point.

Panel (b) of Fig. A1 showcases an example of a iteratively adjusted amplitude adjusted Fourier transform (IAAFT) surrogate (Schreiber and Schmitz, 1996) obtained from the signal from panel a, and panel d show the results of the Pelt algorithm for different penalty parameters.

Figure A2 displays the same analysis, but now for the Gulf Stream region. In the case of this signal, the surrogates produce change points that are quite robust (see Fig. A2d) which increases the significance threshold (black continuous line) considerably. However, two significant change points are detected, which are also robust.

Finally, Table 1 summarizes the significant change points returned by the PELT algorithm. Here we can see how the 2007 transition is widely detected by different ordinal measures (H and SMI), in both regions and mainly on the small scale ( $\delta=1$ ). However, the other transitions, specifically the ones in 2013 and 2021, are only detected in the Gulf Stream region. Additionally, it is clear that the majority of the transitions are only detected by SMI, computed from ordinal pattern probabilities.

Table 1. Summary of the significant change points detected by the PELT algorithm. An asterisk next to the R value indicates that the change point is above the robustness threshold.

| Year | Measure                                 | Region      | Figure | R      |
|------|-----------------------------------------|-------------|--------|--------|
| 1982 | $H_{WE}(ERA5)$ with $\delta = 1$        | Niño3.4     | 3b     | 13.7   |
| 1992 | $H_{NS}(NOAA)$ with $\delta = 8$        | Niño3.4     | 4a     | 5.50   |
| 1997 | $H_{NS}(NOAA)$ with $\delta = 8$        | Niño3.4     | 4a     | 5.50   |
| 1000 | $H_{NS}(NOAA)$ with $\delta = 1$        | Gulf Stream | 3с     | 8.00   |
| 1999 | $H_{NS}(NOAA)$ with $\delta = 8$        | Niño3.4     | 4a     | 5.50   |
|      | $H_{WE}(\text{ERA5})$ with $\delta = 1$ | Niño3.4     | 3b     | 51.7*  |
|      | $H_{NS}(\text{ERA5})$ with $\delta = 1$ | Gulf Stream | 3с     | 19.0*  |
|      | $H_{WE}(\text{ERA5})$ with $\delta = 1$ | Gulf Stream | 3d     | 61.0*  |
|      | $SMI_{WE}$ with $\delta=1$              | Niño3.4     | 7a     | 111.5* |
|      | $SMI_{WE}$ with $\delta=1$              | Gulf Stream | 7b     | 39.3*  |
|      | $SMI_{NS}$ with $\delta=1$              | Niño3.4     | 7c     | 95.5*  |
| 2007 | $SMI_{NS}$ with $\delta=1$              | Gulf Stream | 7d     | 45.8*  |
|      | $SMI_{WE}$ with $\delta = 8$            | Niño3.4     | 7e     | 72.3*  |
|      | $SMI_{WE}$ with $\delta = 8$            | Gulf Stream | 7f     | 305*   |
|      | $SMI_{NS}$ with $\delta = 8$            | Niño3.4     | 7g     | 96.0*  |
|      | $SMI_{NS}$ with $\delta = 8$            | Gulf Stream | 7h     | 104*   |
|      | $SMI_{hist}$                            | Niño3.4     | 7i     | 14.5   |
|      | $SMI_{hist}$                            | Gulf Stream | 7j     | 176*   |
| 2008 | $H_{NS}(NOAA)$ with $\delta = 1$        | Gulf Stream | 3c     | 8.00   |
|      | $H_{NS}(\text{ERA5})$ with $\delta = 1$ | Gulf Stream | 3c     | 19.0*  |
| 2012 | $H_{WE}(\text{ERA5})$ with $\delta = 1$ | Gulf Stream | 3d     | 61.0*  |
| 2013 | $SMI_{WE}$ with $\delta = 1$            | Gulf Stream | 7b     | 7.27   |
|      | $SMI_{NS}$ with $\delta=1$              | Gulf Stream | 7d     | 13.7   |
|      | $SMI_{NS}$ with $\delta=1$              | Niño3.4     | 7c     | 29*    |
| 2015 | $SMI_{hist}$                            | Niño3.4     | 7i     | 14.5   |
|      | r                                       | Niño3.4     | 8c     | 9.9    |
| 2016 | r                                       | Gulf Stream | 8d     | 12.2   |
| 2016 | $SMI_{WE}$ with $\delta = 8$            | Gulf Stream | 7f     | 66*    |
|      | $SMI_{WE}$ with $\delta = 1$            | Gulf Stream | 7b     | 7.27   |
|      | $SMI_{NS}$ with $\delta=1$              | Gulf Stream | 7d     | 10.0   |
| 2021 | $SMI_{WE}$ with $\delta = 8$            | Gulf Stream | 7f     | 20*    |
|      | $SMI_{NS}$ with $\delta = 8$            | Gulf Stream | 7h     | 23*    |
|      | $SMI_{hist}$                            | Gulf Stream | 7j     | 41.0*  |
| 2022 | $H_{WE}$ (NOAA) with $\delta = 1$       | Gulf Stream | 3d     | 8.00   |

Author contributions. JG: conceptualization, formal analysis, methodology, software, writing – original draft, writing – review & editing. 370 GT: conceptualization, software, supervision, writing – original draft, writing – review & editing. CM: conceptualization, supervision, writing – original draft, writing – review & editing. MB: supervision, writing – original draft, writing – review & editing.

Competing interests. The authors declare that they have no conflict of interest.

Acknowledgements. We acknowledge the support of ICREA ACADEMIA, AGAUR (2021 SGR 00606 and FI scholarship), and Ministerio de Ciencia e Innovación (Project No. PID2024-160573NB-I00).

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
