# Peer review of "Detecting transitions and quantifying differences in two SST datasets using spatial permutation entropy"

_EGUsphere, 2025_

## Referee Comment (RC1)

I have reviewed the previous version of this paper, and I find the revised manuscript well written, and much clearer and easier to follow. The analysis is now better articulated and presented with greater detail. I believe the work is ready for publication after some small **clarifications and minor edits**. My comments below are minor and mostly high-level, aimed at clarifying a few parts of the text.

**Comments.**

**- Introduction.**

- The authors should acknowledge the existence of other entropy quantifier for time series: e.g. the weighted permutation entropy by Fadlallah et al. (2013) (see <a href="https://journals.aps.org/pre/abstract/10.1103/PhysRevE.87.022911">https://journals.aps.org/pre/abstract/10.1103/PhysRevE.87.022911</a>) as well as the work of Corso et al. (2020) (see <a href="https://pubs.aip.org/aip/cha/article/30/4/043123/211455/Maximum-entropy-principle-in-recurrence-plot">https://pubs.aip.org/aip/cha/article/30/4/043123/211455/Maximum-entropy-principle-in-recurrence-plot</a>).
- O I also think that the weighted permutation entropy introduced by Fadlallah et al. (2013) could be especially relevant to mention in the conclusion as a possible direction for future work. Specifically, given the spatial permutation entropy proposed in this manuscript, is it feasible to introduce weights to the spatial ordinal patterns in an analogous way to how Fadlallah et al. weighted temporal ordinal patterns? A "spatial weighted permutation entropy" could be an interesting generalization and may be worth briefly discussing as a potential extension.

**- Section 2: Data.**

- The authors consider monthly SST anomalies. I suspect that these are anomalies to respect to the seasonal cycle, but this is not stated (apologies if I missed this).
  Please confirm that this is the case and add it in the paper.
- o The authors chose two regions: the El Niño3.4 and the Gulf Stream. While both regions are of clear importance to the climate modeling community, I think it is valuable to add a few sentences clarifying why such regions were chosen for the general reader.

**- Section 3: Analysis tools.**

- o By reading Bandt and Pompe (2002), it appears that the parameter L is connected to the embedding dimension of the underlying dynamical system. In the context of the Spatial Permutation Entropy introduced in this manuscript, is there any analogous physical interpretation for the parameters L and  $\delta$ ? It may be that these parameters primarily have a statistical role. However, if a physical interpretation exists, it would be helpful to discuss this in this section.
- O Related to above: is there a physical reason to use L = 4? I understand that exploring all possible values of L is beyond the scope of this work, and the results presented are already compelling with this choice, but it would be helpful to clarify if there is a general guideline to choose L.

**- Section 4: Results**

- In Line 145: "To objectively quantify..." I appreciate that the authors used the PELT algorithm to identify shifts in entropy. However, many of the changes highlighted later in the manuscript are already visible by eye (which is positive and further reflect the utility of the spatial permutation entropy metric). It would be helpful to clarify (perhaps in this Section?) that while the PELT algorithm is a valuable and systematic tool, especially for future studies that may extend this analysis to many more regions, the main features in the present results are sufficiently clear to be identified through simple visual inspection. In other words, visual inspection provides a first-order confirmation of shifts in entropy, and the PELT algorithm serves as a helpful, complementary method that could be also further refined or expanded in future work.
- o Line 167. The fact that changes can be sometimes identified in the H\_{WE} direction but not in the H\_{NS} direction in the Niño3.4 region ppears physically meaningful. I suggest the authors briefly highlight that ENSO is a dominant mode characterized by large zonal (rather than meridional) temperature gradient changes, and that the spatial permutation entropy is therefore most sensitive in the direction of the largest gradients. This would help the reader connect the directional differences in entropy changes to known physical mechanisms.
- Related to what asked above in "Section 3: Analysis tools". It appears that \delta clearly carries some physical meaning rather than just statistical. Smaller \delta allow to characterize changes at small spatial scales while larger \delta are more linked to larger, homogeneous changes driven by climate change. The two reanalyses then differ in terms of small spatial variability while capturing the same large scale warming signal.
  - It would be then useful to briefly describe this when introducing the spatial entropy tool. This could also be important to clarify that different \delta allows us to quantify different aspects of the dynamics: some researchers may be more interested in quantifying large scale differences rather than small scale variability.
- Section 4.2: Line 207. The two datasets should indeed capture the same large scale global warming signal, therefore leading to small differences in H\_{WE} and H\_{NS}. If this is indeed the case I would add a small comment. Something of this kind perhaps: "...which indicate that the differences found between ERA5 and NOAA occur mainly at short time scales and that the large scale, low-frequency warming signals are correctly identified in both."
- O Given the comments above, I suggest the following analysis: it would be interesting to examine how the spatial entropy of temperature anomalies changes after removing a linear trend from the raw data. I am curious whether, in this case, the results obtained with small and large  $\delta$  become more similar. If the authors prefer not to perform this analysis, please provide a justification.
- o Figures 6 and 7 are somewhat hard to follow. I suggest adding, on each panel, an indication of whether it corresponds to the Niño3.4 or Gulf Stream region. This would make the figures much clearer. I understand that this information is

described in the caption, but given the large number of panels, providing this clarification directly on each panel would greatly improve readability.

- 4.3. Summary and robustness of detected points.
  - Line 255, "Wald test": please add a citation.
  - I feel this section interrupts the flow of the paper. I would suggest moving it to an appendix, though the authors may choose to retain it in the main text if they prefer.

**- Section 5: Conclusions**

There is currently strong interest in the climate community in developing neural climate emulators. Recent work has highlighted the limitations of such tools, aiming to motivate improved neural network architectures or data-driven strategies (see, e.g., <a href="https://arxiv.org/abs/2510.04466">https://arxiv.org/abs/2510.04466</a>). I think the proposed method could be further adopted as a novel metric to explore discrepancies in emerging AI models. The authors could add one sentence on this as a potential direction for future work.

---

## Author Comment (AC1)

**Authors' response to Anonymous Referee #1**

***Referee comment:*** *I have reviewed the previous version of this paper, and I find the revised manuscript well written, and much clearer and easier to follow. The analysis is now better articulated and presented with greater detail. I believe the work is ready for publication after some small clarifications and minor edits. My comments below are minor and mostly high-level, aimed at clarifying a few parts of the text.*

**Authors' response:** We thank the reviewer for his/her positive opinion and the comments that can allow us to improve our work.

***Referee comment:*** *Introduction.The authors should acknowledge the existence of other entropy quantifier for time series: e.g. the weighted permutation entropy by Fadlallah et al. (2013) (see https://journals.aps.org/pre/abstract/10.1103/PhysRevE.87.022911) as well as the work of Corso et al. (2020) (see https://pubs.aip.org/aip/cha/article/30/4/043123/211455/Maximum-entropy-principle-in-recurrence-plot).*

**Authors' response:** We agree with the reviewer that there are other entropy quantifiers and in the revised manuscript we will include a paragraph about this point, and cite the reference suggested by the reviewer, as well as other works.

***Referee comment:*** *I also think that the weighted permutation entropy introduced by Fadlallah et al. (2013) could be especially relevant to mention in the conclusion as a possible direction for future work. Specifically, given the spatial permutation entropy proposed in this manuscript, is it feasible to introduce weights to the spatial ordinal patterns in an analogous way to how Fadlallah et al. weighted temporal ordinal patterns? A "spatial weighted permutation entropy" could be an interesting generalization and may be worth briefly discussing as a potential extension.*

**Authors' response:** We agree with the reviewer that a spatial extension of Fadlallah et al. "weighted permutation entropy" can yield interesting results and is a natural continuation of the present work. We will include a comment in the conclusions.

***Referee comment:*** *Section 2: Data.The authors consider monthly SST anomalies. I suspect that these are anomalies to respect to the seasonal cycle, but this is not stated (apologies if I missed this). Please confirm that this is the case and add it in the paper.*

**Authors' response:** Yes, we analyze anomalies with respect to the seasonal cycle, and we will clarify this in the revised manuscript.

***Referee comment:*** *The authors chose two regions: the El Niño3.4 and the Gulf Stream. While both regions are of clear importance to the climate modeling community, I think it is valuable to add a few sentences clarifying why such regions were chosen for the general reader.*

**Authors' response:** We agree with the reviewer and in the revised manuscript we will include a sentence explaining that these regions were chosen not only because they are of clear importance, but also because they have different spatio-temporal SST dynamics. SST in El Niño region is governed by tropical dynamics, and in particular the SST dynamics results from ocean-atmosphere interactions leading to variability mainly on interannual time scales. On the other hand, the Gulf Stream dynamics, as one of the most intense western boundary currents, is governed by internal ocean dynamics and the extratropical winds across the basin, resulting in SST variability on several time scales, from fast changes due to atmospheric-driven heat fluxes to decadal shifts in spatial structure.

***Referee comment:*** *Section 3: Analysis tools. By reading Bandt and Pompe (2002), it appears that the parameter $L$ is connected to the embedding dimension of the underlying dynamical system. In the context of the Spatial Permutation Entropy introduced in this manuscript, is there any analogous physical interpretation for the parameters $L$ and $\delta$? It may be that these parameters primarily have a statisticarole. However, if a physical interpretation exists, it would be helpful to discuss this in this section.*

**Authors' response:** Indeed, in the original application of permutation entropy to time series analysis, L and $\delta$ allowed to embed the time series in a L-dimensional space. In the context of the spatial approach, in our opinion these parameters play a similar role: by tunning the spatial scale covered by the ordinal pattern (in the horizontal or in the vertical orientation) they allow to embed a set of gridded datapoints to obtain an instantaneous ``snapshot'' of a two-dimensional field in a low-dimensional space. We'll mention this point in the revised manuscript.

***Referee comment:*** *Related to above: is there a physical reason to use L = 4? I understand that exploring all possible values of $L$ is beyond the scope of this work, and the results presented are already compelling with this choice, but it would be helpful to clarify if there is a general guideline to choose L.*

**Authors' response:** To the best of our knowledge, there is no general guideline to choose L, except for the limitation of having enough datapoints to define a sufficiently large number of ordinal patterns, in order to have good statistics to estimate the L! probabilities of the possible ordinal patterns. We used L=4 as a compromise to analyze long correlations with good statistics. However, we checked that L=3 and L=5 produced similar results (see Figs. 1 and 2) and in the revised manuscript we will include a sentence clarifying this point.

[Figure]

Fig. 1: Analysis of El Niño region, using ordinal patterns of length L=3 (top row), L=4 (as in the manuscript, middle row) and L=5 (bottom row). The two columns correspond to the two orientations of the ordinal patterns (left: NS; right: WE).

[Figure]

Fig. 2: as Fig. 1 but for El Gulf region.

***Referee comment:*** *Section 4: Results. In Line 145: "To objectively quantify…" I appreciate that the authors used the PELT algorithm to identify shifts in entropy. However, many of the changes highlighted later in the manuscript are already visible by eye (which is positive and further reflect the utility of the spatial permutation entropy metric). It would be helpful to clarify (perhaps in this Section?) that while the PELT algorithm is a valuable and systematic tool, especially for future studies that may extend this analysis to many more regions, the main features in the present results are sufficiently clear to be identified through simple visual inspection. In other words, visual inspection provides a first-order confirmation of shifts in entropy, and the PELT algorithm serves as a helpful, complementary method that could be also further refined or expanded in future work.*

**Authors' response:** We agree with the referee that the main shifts in entropy are clearly identified by visual inspection, and PELT provides a complementary identification. We will clarify this aspect in the revised manuscript. We also agree that PELT performance should be, for future work, refined and better understood.

***Referee comment:*** *Line 167. The fact that changes can be sometimes identified in the $H_{WE}$ direction but not in the $H_{NS}$ direction in the Niño3.4 region ppears physically meaningful. I suggest the authors briefly highlight that ENSO is a dominant mode characterized by large zonal (rather than meridional) temperature gradient changes, and that the spatial permutation entropy is therefore most sensitive in the direction of the largest gradients. This would help the reader connect the directional differences in entropy changes to known physical mechanisms.*

**Authors' response:** We really appreciate this comment of the reviewer that help us interpret the reason why, in the Niño3.4 region, some changes are identified in $H_{WE}$ but not in $H_{NS}$. In the revised manuscript we will include a comment about this point.

***Referee comment:*** *Related to what asked above in "Section 3: Analysis tools". It appears that \delta clearly carries some physical meaning rather than just statistical. Smaller \delta allow to characterize changes at small spatial scales while larger \delta are more linked to larger, homogeneous changes driven by climate change. The two reanalyses then differ in terms of small spatial variability while capturing the same large scale warming signal.*

**Authors' response:** Yes, indeed, $\delta$ allows to tune the spatial scale covered by the ordinal pattern and our results show that the two reanalysis products differ in short spatial scales while are consistent in the long scales. In the revised manuscript we will modify the conclusions to more clearly explain this idea.

***Referee comment:*** *It would be then useful to briefly describe this when introducing the spatial entropy tool. This could also be important to clarify that different \delta allows us to quantify different aspects of the dynamics: some researchers may be more interested in quantifying large scale differences rather than small scale variability.*

**Authors' response:** We agree with the reviewer and in the introduction, we will stress that a main advantage of the ordinal methodology for the analysis of a climatological dataset (or for the comparison of two datasets) is the possibility of selecting the spatial lag $\delta$ according to the interest of the research: to capture changes (or to identify differences) in short or in long spatial scales.

***Referee comment:*** *Section 4.2: Line 207. The two datasets should indeed capture the same large scale global warming signal, therefore leading to small differences in $H_{WE}$ and $H_{NS}$. If this is indeed the case I would add a small comment. Something of this kind perhaps: "…which indicate that the differences found between ERA5 and NOAA*

*occur mainly at short time scales and that the large scale, low-frequency warming signals are correctly identified in both."*

**Authors' response:** We thank the reviewer for this comment. We will modify the text to clarify that differences between ERA5 and NOAA occur at short time and spatial scales, while at long scales, the warming signals are consistent in both datasets.

*Referee comment: Given the comments above, I suggest the following analysis: it would be interesting to examine how the spatial entropy of temperature anomalies changes after removing a linear trend from the raw data. I am curious whether, in this case, the results obtained with small and large $\delta$ become more similar. If the authors prefer not to perform this analysis, please provide a justification.*

**Authors' response:** The analysis was performed after removing the seasonal cycle but not the linear trend. As it can be seen in Figs. 3-6, removing the linear trend has almost no effect in the agreement between the datasets, but reduces the trends in SPE for $\delta=8$. We also performed the analysis on the "raw data" (without removing the seasonal cycle) and as shown in Figs. 3-6, depending on the region and method used, it can return additional information. Specifically, the variation of the entropies in the top row of Figs. 3 and 4 is consistent with the fact that in the equatorial Pacific, the seasonal cycle is significant in the WE direction and therefore manifests itself on both short and long scales. In the Gulf Stream, Figs. 5 and 6, the seasonal cycle is more significant in the NS direction because the current is nearly zonal. Therefore, it can be seen in NS when using $\delta=1$ (panel a in Fig. 5), but not in the WE direction (panel b in Fig. 5). However, when $\delta=8$, the distances are large enough for the seasonal cycle to also manifest itself also in the WE direction (panel b in Fig. 6). In the revised manuscript we plan to include a comment about this point.

[Figure]

Fig. 3: Analysis of El Niño region, with a small spatial lag ($\delta=1$) of the raw data (top row), of the anomalies (after removing the seasonal cycle, as in the manuscript, middle row) and of the detrended anomalies (after removing the linear trend and the seasonal cycle, bottom row). The columns display results for the two OP orientations (left: NS; right: WE).

[Figure]

Fig. 4: as Fig. 3 but for $\delta=8$.

[Figure]

Fig. 5: as Fig. 3 (δ=1) but for El Gulf region.

[Figure]

Fig. 6: as Fig. 4 (δ=8) but for El Gulf region.

*Referee comment: Figures 6 and 7 are somewhat hard to follow. I suggest adding, on each panel, an indication of whether it corresponds to the Niño3.4 or Gulf Stream region. This would make the figures much clearer. I understand that this information is described in the caption, but given the large number of panels, providing this clarification directly on each panel would greatly improve readability.*

**Authors' response:** We agree with the reviewer that this modification will improve readability and in the revised manuscript we will modify these figures accordingly.

*Referee comment: 4.3. Summary and robustness of detected points. Line 255, "Wald test": please add a citation.*

**Authors' response:** In the revised manuscript we will add the following citation "Fahrmeir, Ludwig; Kneib, Thomas; Lang, Stefan; Marx, Brian (2013). Regression: Models, Methods and Applications" section 5.1.3)."

*Referee comment: I feel this section interrupts the flow of the paper. I would suggest moving it to an appendix, though the authors may choose to retain it in the main text if they prefer.*

**Authors' response:** In the revised manuscript we will move the section to an appendix.

*Referee comment: Section 5: Conclusions. There is currently strong interest in the climate community in developing neural climate emulators. Recent work has highlighted the limitations of such tools, aiming to motivate improved neural network architectures or data-driven strategies (see, e.g., https://arxiv.org/abs/2510.04466). I think the proposed method could be further adopted as a novel metric to explore discrepancies in emerging AI models. The authors could add one sentence on this as a potential direction for future work.*

**Authors' response:** We again thank the reviewer for this comment. Indeed, a very useful application of the ordinal method will be to identify discrepancies in AI models. We will mention this in the conclusions as a possible direction of research.